# Lipotoxicity in a Vicious Cycle of Pancreatic Beta Cell Exhaustion

**DOI:** 10.3390/biomedicines10071627

**Published:** 2022-07-07

**Authors:** Vladimir Grubelnik, Jan Zmazek, Matej Završnik, Marko Marhl

**Affiliations:** 1Institute of Mathematics and Physics, Faculty of Electrical Engineering and Computer Science, University of Maribor, 2000 Maribor, Slovenia; vlado.grubelnik@um.si; 2Department of Physics, Faculty of Natural Sciences and Mathematics, University of Maribor, 2000 Maribor, Slovenia; jan.zmazek@um.si; 3Department of Endocrinology and Diabetology, University Medical Center Maribor, Ljubljanska ulica 5, 2000 Maribor, Slovenia; matej.zavrsnik1@gmail.com; 4Department of Elementary Education, Faculty of Education, University of Maribor, 2000 Maribor, Slovenia; 5Department of Biophysics, Faculty of Medicine, University of Maribor, 2000 Maribor, Slovenia

**Keywords:** diabetes, insulin secretion, lipids, PEP cycle, uncoupling proteins, mitochondrial dysfunction

## Abstract

Hyperlipidemia is a common metabolic disorder in modern society and may precede hyperglycemia and diabetes by several years. Exactly how disorders of lipid and glucose metabolism are related is still a mystery in many respects. We analyze the effects of hyperlipidemia, particularly free fatty acids, on pancreatic beta cells and insulin secretion. We have developed a computational model to quantitatively estimate the effects of specific metabolic pathways on insulin secretion and to assess the effects of short- and long-term exposure of beta cells to elevated concentrations of free fatty acids. We show that the major trigger for insulin secretion is the anaplerotic pathway via the phosphoenolpyruvate cycle, which is affected by free fatty acids via uncoupling protein 2 and proton leak and is particularly destructive in long-term chronic exposure to free fatty acids, leading to increased insulin secretion at low blood glucose and inadequate insulin secretion at high blood glucose. This results in beta cells remaining highly active in the “resting” state at low glucose and being unable to respond to anaplerotic signals at high pyruvate levels, as is the case with high blood glucose. The observed fatty-acid-induced disruption of anaplerotic pathways makes sense in the context of the physiological role of insulin as one of the major anabolic hormones.

## 1. Introduction

Metabolic syndrome is a burdensome public health problem in modern society and is closely related to type 2 diabetes mellitus (T2DM). Obesity and physical inactivity are the two most important risk factors for the development of metabolic syndrome, which is composed of a set of conditions that include high fasting blood glucose and abnormal cholesterol and triglyceride levels, in addition to hypertension and central obesity [1]. Dyslipidemias involving hypertriglyceridemia and low high-density lipoprotein levels are the major phenotypes associated with T2DM [2]. The influence of lipids on glucose metabolism, particularly the role of ectopic lipid accumulation, requires a systemic approach in a complex circuit of the endocrine pancreas and its hormone-targeted tissues, especially the liver and muscle [3,4]. We and others have shown that hyperlipidemia may statistically precede hyperglycemia by several years [5,6], indicating that it is predictive of reduced glucose tolerance in certain individuals. In addition, insulin resistance and prediabetes, conditions closely associated with metabolic syndrome, are thought to be caused by disturbances in energy utilization and storage, which certainly include impaired lipid utilization. While fasting hyperinsulinemia predates hyperglycemia in most cases, suggesting that insulin resistance likely precedes beta cell dysfunction, it is unclear whether fasting insulin hypersecretion is a primary driver of insulin resistance or a consequence in the form of a compensatory whole-body response to insulin resistance [7]. It was also shown that primary insensitivity to insulin does not appear to be fundamental to the pathogenesis of hyperlipidemia in familial dysbetalipoproteinemia [8]. Therefore, it is of great research interest to clarify whether high blood lipids contribute to dysfunctional glucose-stimulated insulin secretion (GSIS).

Several studies have revealed the mechanisms of lipotoxicity on beta cell function in association with T2DM [9,10,11]. Besides their involvement in tissue inflammation and the development of insulin resistance in peripheral tissues, free fatty acids (FFAs), particularly long-chain (Lc-FFAs) and saturated FFAs, might contribute to abnormal insulin response at several stages of their metabolism. These stages include FFA entry, mitochondrial metabolism, degradation [12], or even the triggering of ER stress-induced apoptosis [13]. Palmitate, the most common saturated FFA found in the human body, plays an important role in the lipotoxicity. It represents 20–30% of total FFAs in membrane phospholipids and adipose tissue triacylglycerols (TAGs) [14], and can be obtained in the diet [15] or synthesized endogenously [11]. In contrast, unsaturated FFAs have been assigned protective roles against the lipotoxic effects of saturated FFAs, such as preventing beta-cell apoptosis, regulating plasma glucose concentrations, and enhancing insulin sensitivity [11,16]. Oleic acid is the most abundant unsaturated FFA in human adipose tissue [17], promoting neutral lipid accumulation and insulin secretion. On the other hand, palmitic acid is poorly incorporated into triglycerides and, at physiological glucose concentrations, does not promote insulin secretion from human pancreatic islets [18].

Beta cells express the FFA transporter cluster of differentiation 36 (CD36), also known as fatty acid translocase (FAT) [19], and FFA receptors (FFAR1, FFAR2, and FFAR3). Short-chain FFAs (Sc-FFAs), produced by gut microbiota, target FFAR2 and FFAR3, but their role in beta cell function is unclear [20]. On the other hand, Lc-FFAs primarily stimulate CD36 and FFAR1 transporters. The latter induce basal hypersecretion of insulin secretion [9] by the activation of inositol triphosphate (IP3) and diacylglycerol (DAG) pathways and increasing cytosolic calcium concentration. At the mitochondrial level, FFAs decrease glutamine levels by the deterioration of anaplerosis (by promoting the formation of glutamate from glutamine). Chronic elevation of FFAs contributes to mitochondrial dysfunction and cellular senescence via signaling pathways and increased oxidative stress levels [12]. Moreover, several studies have linked chronically elevated FFAs levels to increased uncoupling protein 2 (UCP2) expression [21,22,23,24], typically observed in T2DM patients. The role of UCP2 in insulin secretion has also been recently reviewed [25], describing studies linking UCP2 to obesity that focus on the inflammatory process associated with ROS. The physiological significance of high UCP2 expression is unclear, but it might represent a signal for beta cells to prefer FFA and amino acid oxidation instead of glucose [24,26,27,28,29,30]. Lastly, the degradation of FFAs leads to ceramide accumulation and ceramide-associated beta-cell dysfunction [31].

In general, acute exposure of beta cells to FFAs has been recognized as beneficial for GSIS. In contrast, chronic increases in plasma FFA concentrations led to disruptions in the regulation of lipid metabolism and impaired beta-cell function due to lipotoxic effects [10]. Moreover, the adaptive signaling pathways induced to counteract lipotoxic stress have secondary adverse effects, as antilipotoxic signaling cascades may contribute to beta cell failure [32]. Experimental results from a recent study suggest a novel non-esterified FFA–stimulated pathway that selectively drives pancreatic islet non-glucose-stimulated insulin secretion (NGSIS) [7]. At low glucose levels reflecting a fasting state, FFAs affect NGSIS by inducing an H^+^ leak at the inner mitochondrial membrane that drives tricarboxylic acid (TCA) flux to maintain mitochondrial membrane potential. Combining this knowledge with the recent discovery about the role of the phosphoenolpyruvate (PEP) cycle in beta cells and insulin secretion [33], the increased TCA cycle flux corresponds to the increased PEP cycle and NGSIS. The effect of the PEP cycle on insulin secretion has been recently reviewed [34], summarizing advances in understanding the metabolic mechanisms involved in insulin secretion.

Here we present a computational model including the anaplerotic and cataplerotic pathways of GSIS, which aims to explain the effects of acutely and chronically elevated FFAs on GSIS. Anaplerosis has been recognized as an essential pathway implicated in beta-cell activation by glucose already in early studies employing NMR carbon isotope tracing [35], which was later recognized as one of the significant pathways promoting insulin secretion [36]. Recently, a new conceptual framework for GSIS has been proposed based on the specific role of mitochondrial metabolism and the PEP cycle [33], disrupting the established “consensus model” [37]. The PEP cycle assumes the upregulation of anaplerotic pathways induced by the high expression of the pyruvate carboxylase (PC) enzyme in beta cells [38]. Instead of entering the oxidative metabolism by conversion to acetyl-CoA via pyruvate dehydrogenase (PDH), pyruvate flux is diverted to the production of oxaloacetate (OAA), an intermediate of the TCA cycle. OAA is converted to PEP, which leaves the mitochondria, diffuses, and accumulates locally at the plasma membrane. The conversion of PEP to pyruvate catalyzed by membrane-bound pyruvate kinase (PK) increases the cytosolic ATP/ADP ratio to the level required to inhibit the K_ATP_ channel [33,39]. The vital signaling role of anaplerosis and the PEP cycle for glucose sensing and insulin secretion is confirmed by several experimental studies. PC expression in beta cells is higher than in other tissues [35], and the anaplerotic flux from pyruvate to OAA via PC is strongly responsive to changes in extracellular glucose [40]. In fact, a study on the rat INS-1 insulinoma cell line showed a stronger correlation of GSIS with anaplerotic than with oxidative metabolism of pyruvate [41]. In this context, we also discuss the role of UCP2, which blocks the oxidation of pyruvate and paves the way for its anaplerotic fate [42]. The importance of the PEP-cycle in GSIS is further supported by the results showing that an enhanced TCA-cycle-derived mitochondrial GTP (mGTP) turnover amplifies insulin secretion by increasing insulin content, granule docking, and mitochondrial mass [43,44]. In addition, pyruvate kinase activators (PKa) amplify insulin release via the PEP cycle in preclinical T2DM models [45].

In the following sections, we first present a model of beta cell function, followed by the results in terms of model predictions for insulin secretion at low and high glucose levels. The model predictions agree well with the known mechanisms of dysregulations of insulin secretion, i.e., excessively high fasting levels and inadequate postprandial insulin secretion caused by the pathophysiological conditions of hyperlipidemia, and contribute to a better understanding of the general dysregulations of insulin signaling in obesity and T2DM.

## 2. Materials and Methods

The model considers the major metabolic pathways of glucose and FFAs that trigger insulin exocytosis. The metabolic pathways of glucose, the metabolism of FFA, and the mechanism of insulin secretion are shown schematically in Figure 1. The importance of the TCA and PEP cycles in mitochondria is particularly emphasized. The PEP cycle and transport of metabolites to microdomains at the plasma membrane cause an increase in ATP concentration near K_ATP_ channels, triggering insulin secretion. Glucose and FFAs enter the cell via glucose transporters (GLUTs) and FFA transporters (e.g., CD36), where they are first metabolized by glycolytic and beta-oxidation pathways, respectively. Glucose-derived pyruvate enters either the cataplerotic direction of the TCA cycle (by conversion to acetyl-CoA) or the anaplerotic direction (by conversion to OAA). Beta-oxidation-derived acetyl-CoA can only enter the cataplerotic metabolism and accelerate the TCA cycle. The increased citrate concentrations induce the formation of malonyl-CoA (MaCoA) in the cytosol and inhibit the entry of fatty acyl-coenzyme A (FA-CoA) via carnitine palmitoyltransferase 1 (CPT1). The mGTP and PEP cycles, which occur concurrently with the TCA cycle, convert OAA to PEP, which is exported from mitochondria and converted back to pyruvate in the cytosol, resulting in ATP production in the K_ATP_ channel microdomains (indicated by the red dashed line in Figure 1). This microdomain ATP concentration also depends on the “classical” energy generation pathway, which consists of the glucose and FFA oxidation pathways. NAD(P)H and FADH_2_ are oxidized by the electron transport chain (ETC), generating ATP molecules and increasing the global cytosolic ATP concentration (ATP_cyt_) that can diffuse into/from microdomains. The ATP-dependent closure of K_ATP_ channels leads to membrane potential oscillations and the influx of Ca^2+^ ions through voltage-gated calcium channels (VGCC). Increased Ca^2+^ concentrations induce the Ca^2+^-dependent exocytosis of insulin granules. The mathematical modeling of the processes is described in detail in the following subsections.

### 2.1. Glucose Metabolism

Glucose is first transported into the intracellular space and phosphorylated to glucose-6-phosphate (G6P) during the first priming reaction of glycolysis. This process is considered the rate-limiting step of glycolysis and regulates the glycolytic flux. The uptake of glucose and its phosphorylation are modeled by Michaelis-Menten kinetics (see [46]):(1)$$J_{G6P}=J_{\max}\left(1-k_{\mathrm{FFA}}f_{\mathrm{FFA}}\right)\frac{G^{2}}{K_{m}^{2}+G^{2}}$$

The parameters $J_{\max}=8\;\mu M/s\;$and $K_{m}=3.8\;\mathrm{mM}$ were chosen based on experimental data for glucose concentrations *G* at 1 mM and 10 mM [35]. Equation (1) also considers the inhibition of glycolytic flux caused by FFAs, where the parameter $f_{\mathrm{FFA}}$ represents the exposure to FFA. Based on the experimental results of Roden [47], it is assumed that acyl-CoA blocks glucose phosphorylation. The value of $k_{\mathrm{FFA}}=0.08$ is chosen such that $J_{G6P}$ decreases by about 20% when $f_{\mathrm{FFA}}$ is increased from 0 to 2.5. This is consistent with the results of Roden, who found that glycolytic flux can be decreased by more than 20% when the plasma concentration of FFA is doubled [47].

In the fourth step of glycolysis, the 6-carbon molecule splits into two 3-carbon molecules, so the glycolytic flux is doubled, which is modeled as follows:(2)$$J_{\mathrm{gly}}=2\;J_{G6P}.$$

The final products of glycolysis are two molecules of pyruvate. Considering the preparatory and final phases, glycolysis yields a net total of 2 molecules of ATP:(3)$$J_{\mathrm{ATP},\mathrm{gly}}=2\;J_{G6P},\;$$
where $J_{\mathrm{ATP},\mathrm{gly}}$ is the net glycolytic ATP production flux. Glycolysis also yields 2 molecules of NADH. Depending on the amount of LDH expressed in the cell, a portion of pyruvate is reduced to lactate in the reaction in which 1 NADH molecule is oxidized to NAD^+^. We model this effect by introducing a parameter *p*_L_:(4)$$J_{\mathrm{lac}}=p_{L}\;J_{\mathrm{gly}}.$$

Considering that beta cells express very low levels of LDH, we set the parameter to $p_{L}=0.05$ [46]. Thus, the NADH flux generated by glycolysis is:(5)$$J_{\mathrm{NADH},\mathrm{gly}}=\left(1-p_{L}\right)\;J_{\mathrm{gly}}.\;$$

To the same extent, a fraction of pyruvate generated by glycolysis that is not reduced to lactate enters the mitochondria:(6)$$J_{\mathrm{pyr}}=\left(1-p_{L}\right)\;J_{\mathrm{gly}}.$$

The transport of pyruvate into the mitochondria ($J_{\mathrm{pyr}}$), the oxidation of pyruvate to acetyl-CoA, and the production of reducing equivalents in the TCA cycle are modeled by the following equation:(7)$$J_{\mathrm{pyr},\mathrm{TCA}}=\left(1-f_{\mathrm{ANA}}\right)J_{\mathrm{pyr}}.\;$$

A portion of pyruvate ($f_{\mathrm{ANA}}$) is used for anaplerotic reactions, which represents an important part of beta cell metabolism related to insulin secretion. In this process, FFAs play an important role by additionally diverting pyruvate flux to anaplerosis. This is particularly important in long-term and chronic hyperlipidemia and may be related, at least in part, to the increase in UCP2, which serves as a metabolic switch that prevents pyruvate oxidation and saves it for anabolic purposes [24,26,27,28,29,30]. This effect of FFAs is modeled by:(8)$$f_{\mathrm{ANA}}=k_{\mathrm{ANA},0}+k_{\mathrm{ANA}}\;f_{\mathrm{FFA}}.$$

Previous experimental data suggest that anaplerosis in beta cells is relatively important, but there is a large discrepancy between the quantitative results of experimental studies [35,48,49]. We set the basal rate of anaplerosis (in the absence of FFAs) to 40% ($k_{\mathrm{ANA},0}=0.4$), which is consistent with the above studies, indicating that approximately 60% of pyruvate is directly oxidized via PDH, while the remaining 40% is carboxylated by PC. The value of $k_{\mathrm{ANA}}=0.08$, which corresponds to the FFA-dependent increase of anaplerosis that is set to mirror a 20% elevation of anaplerosis during full exposure to FFAs (at $f_{\mathrm{FFA}}=2.5$). The maximal anaplerotic flux of 60% was used in a previous study which did not consider variation in FFA exposure [46].

The combined PDH and TCA cycle reactions yield 4 NADH ($J_{\mathrm{NADH},\mathrm{pyr},\mathrm{TCA}}=4J_{\mathrm{pyr},\mathrm{TCA}}$), 1 FADH_2_ ($J_{{\mathrm{FADH}}_{2},\mathrm{pyr},\mathrm{TCA}}=J_{\mathrm{pyr},\mathrm{TCA}}$), and 1 GTP ($J_{\mathrm{GTP},\mathrm{pyr},\mathrm{TCA}}=J_{\mathrm{pyr},\mathrm{TCA}}$) molecule. As described, mitochondrial PEPCK hydrolyzes GTP to produce PEP (see Figure 1). Consequently, the GTP produced by TCA cycle does not contribute to the overall energy production, since it is coupled to the PEP cycle. The coupling of GTP and PEP cycles is modeled in the continuation.

It is generally accepted that oxidation of 1 FADH_2_ by the ETC yields 1.5 ATP, while oxidation of 1 NADH yields 2.5 ATP, commonly referred to as the P/O ratio:(9)$$J_{\mathrm{ATP},\mathrm{pyr},\mathrm{ox}}=\left(2.5\;J_{\mathrm{NADH},\;\mathrm{pyr},\mathrm{TCA}}+1.5\;J_{{\mathrm{FADH}}_{2},\;\mathrm{pyr},\mathrm{TCA}}\right)\left(1-f_{p,\mathrm{leak}}\right)\left(1-k_{\mathrm{md}}\right).$$

Similarly, the glycolysis-produced NADH molecules (Equation (5)) enter mitochondria and the ETC via glycerol-phosphate and malate-aspartate shuttles [50,51], contributing to aerobic ATP production. While the malate-aspartate shuttle transfers NADH molecules directly, the glycerol-phosphate shuttle transports electrons from NADH (by regenerating NAD^+^) to glycerol-3-phosphate dehydrogenase 2, reducing enzyme-bound FAD to FADH_2_. Since beta cells’ GSIS depends on the glycerol-phosphate shuttle [52], we assume the P/O ratio of 1.5. Accordingly, the rate of ATP production from the glycolysis-produced NADH molecules is given by:(10)$$J_{\mathrm{ATP},\mathrm{NADH},\mathrm{gly},\mathrm{ox}}=1.5\;J_{\mathrm{NADH},\mathrm{gly}}\left(1-f_{p,\mathrm{leak}}\right)\left(1-k_{\mathrm{md}}\right).$$

Thus, the total production of ATP by glucose metabolism is:(11)$$J_{\mathrm{ATP},G}=J_{\mathrm{ATP},\mathrm{gly}}+J_{\mathrm{ATP},\mathrm{GO}}=J_{\mathrm{ATP},\mathrm{gly}}+J_{\mathrm{ATP},\mathrm{pyr},\mathrm{ox}}+J_{\mathrm{ATP},\mathrm{NADH},\mathrm{gly},\mathrm{ox}}.$$
ln Equations (9) and (10), we also consider decreasing ATP production due to increased mitochondrial proton leak ($f_{p,\mathrm{leak}}$) and mitochondrial dysfunction ($k_{\mathrm{md}}$). Insulin secretion has been shown to be associated with increased UCP2 expression and decreased glucose-stimulated ATP/ADP ratio due to increased mitochondrial proton leak [53,54]. Several studies have linked chronic hyperlipidemia to increased UCP2 levels [23,24], which are typically observed in T2DM patients. Mathematical models have also been created to account for an increase in proton leak due to uncoupling protein (UCP) activation by ROS [55]. In response to FFAs, proton leak may also be enhanced by mitochondrial permeability transition pores (mPTP) [7]. In view of these observations, mitochondrial proton leak is modeled as follows:(12)$$f_{p,\mathrm{leak}}=k_{p,\mathrm{leak}}\left(J_{\mathrm{pyr},\mathrm{TCA}}+J_{\mathrm{FFA},\mathrm{TCA}}\right).$$

UCP expression has been shown to alter the ratio of proton leak and proton efflux, leading to a reduction in ATP production of approximately 20% at high glucose concentrations (higher mitochondrial membrane potential) [55]. In Equation (12), this corresponds to the value of the parameter $k_{p,\mathrm{leak}}=0.1\;s/\mu M.$

Mitochondrial dysfunction, commonly observed in T2DM and insulin-resistant individuals [56], is taken into account in the model by changing the $k_{\mathrm{md}}$ parameter in Equations (9) and (10), as has been modeled previously [57,58]. In pancreatic tissue, mitochondrial dysfunction has been identified as one of the major causes for impaired secretory response of β-cells to glucose [59,60]. In T2DM, β-cells contain swollen mitochondria with disrupted cristae [53,54,61] and impaired stimulus-secretion coupling. Mitochondrial oxidative phosphorylation has been shown to decrease by 20–40% in insulin-resistant individuals [56,62].

### 2.2. FFA Metabolism

In the model we focus on Lc-FFAs, which serve as an energy source via beta-oxidation. Food is the major source of lipids and more specifically of poly-unsaturated FFAs. TAGs are split to Lc-FFAs by lipases of the digestive juices releasing, among others, palmitate (C16:0, saturated), oleate (C18:1, mono-unsaturated), and linoleate (C18:2, poly-unsaturated) [20]. Beta cell exposure to Lc-FFAs activates CD36 FFA receptors in the cell membrane which play an important role in the uptake of FFAs and have multiple biological functions that may be important in inflammation and in the development of metabolic diseases, including diabetes [19]. The upregulation of the CD36 transporter in beta-cells increases the uptake of FFAs, resulting in enhanced GSIS and impaired oxidative metabolism [63,64,65]. We model FFA exposure by introducing a parameter $f_{\mathrm{FFA}}$ in Equation (13). It should be noted that an increase in blood glucose level blocks beta oxidation. A rise in blood glucose levels increases glucose-derived pyruvate and the activity of the TCA cycle ($J_{\mathrm{pyr},\mathrm{TCA}}$). This results in the production of citrate that escapes the mitochondria and activates the acetyl-CoA carboxylase to generate cytosolic malonyl-CoA. In turn, malonyl-CoA inhibits the CPT1, blocking the entry of acyl-CoA into mitochondria and disabling beta-oxidation [20]. We model this effect with
(13)$$J_{\mathrm{FFA},\;\beta \;}=J_{\mathrm{FFA},\mathrm{TCA},0}\left(1-\frac{J_{\mathrm{pyr},\mathrm{TCA}}}{k_{m,\mathrm{CPT}}+J_{\mathrm{pyr},\mathrm{TCA}}}\right)f_{\mathrm{FFA},p,\mathrm{leak}}\;f_{\mathrm{FFA}}.$$

Experimental results from a recent study also suggest that elevated circulating FFAs increase proton leak in the mitochondria of beta cells, which drives TCA cycle flux to maintain mitochondrial membrane potential [7,46]. To model this effect, the parameter $f_{\mathrm{FFA},p,\mathrm{leak}}$ in Equation (13) accounts for the increase in $J_{\mathrm{FFA},\;\beta }$ due to the increased mitochondrial proton leak:(14)$$f_{\mathrm{FFA},p,\mathrm{leak}}=\left(1+k_{I,\mathrm{FFA}}k_{p,\mathrm{leak}}\right).$$

The constants $J_{\mathrm{FFA},\mathrm{TCA},0}=0.25\;\mu M/s$, $k_{m,\mathrm{CPT}}=8\;\mu M/s$, and $k_{I,\mathrm{FFA}}=4\;\mu M/s$ are determined by qualitatively obtaining the same ATP production at $f_{\mathrm{FFA}}=1$ as in the previous study in which FFA oxidation was determined from oxygen consumption [46].

Once FFAs enter the mitochondrial matrix, they are repeatedly cleaved during the beta-oxidation pathway, producing acetyl-CoA molecules. Calculations of fluxes in the continuation are performed for 18-carbon oleic acid, the most abundant FFA in human adipose tissue [17]. Beta oxidation of oleic acid requires eight consecutive reactions, yielding nine molecules of acetyl-CoA. Each acetyl-CoA is then converted to succinyl-CoA by carboxylation reaction, yielding four net ATP molecules (due to oxidation of 1 NADH and 1 FADH_2_), yielding a net total of 32 ATP molecules. However, two ATP molecules are lost during the activation of each FFA. Therefore, the ATP yield during beta-oxidation can be given as follows:(15)$$J_{\mathrm{ATP},\mathrm{FFA},\beta ,\mathrm{ox}}=30\;J_{\mathrm{FFA},\beta }\;\left(1-f_{p,\mathrm{leak}}\right)\left(1-k_{\mathrm{md}}\right).$$

Due to the breakdown of oleic acid into nine molecules of acetyl-CoA during the beta-oxidation pathway, the acetyl-CoA flux into mitochondria is multiplied:(16)$$J_{\mathrm{FFA},\mathrm{TCA}}=9\;J_{\mathrm{FFA},\beta }.$$

The TCA cycle yields three molecules of NADH ($J_{\mathrm{NADH},\mathrm{FFA},\mathrm{TCA}}=3J_{\mathrm{FFA},\mathrm{TCA}}$), one molecule of FADH_2_ ($J_{{\mathrm{FADH}}_{2},\mathrm{FFA},\mathrm{TCA}}=J_{\mathrm{FFA},\mathrm{TCA}}$), and one molecule of GTP ($J_{\mathrm{GTP},\mathrm{FFA},\mathrm{TCA}}=J_{\mathrm{FFA},\mathrm{TCA}}$). The latter is in turn directly consumed (as in Equation (10)) by the PEP cycle, yielding the following:(17)$$J_{\mathrm{ATP},\mathrm{FFA},\mathrm{TCA},\mathrm{ox}}=\left(2.5\;J_{\mathrm{NADH},\;\mathrm{FFA},\mathrm{TCA}}+1.5\;J_{{\mathrm{FADH}}_{2},\;\mathrm{FFA},\mathrm{TCA}}\right)\left(1-f_{p,\mathrm{leak}}\right)\left(1-k_{\mathrm{md}}\right)$$

Thus, the total production of ATP by FFA metabolism is:(18)$$J_{\mathrm{ATP},\mathrm{FFAO}}=J_{\mathrm{ATP},\mathrm{FFA},\beta ,\mathrm{ox}}+J_{\mathrm{ATP},\mathrm{FFA},\mathrm{TCA},\mathrm{ox}}.$$

The decrease in ATP production due to increase in proton leak ($f_{p,\mathrm{leak}}$) and mitochondrial dysfunction ($k_{\mathrm{md}}$) in Equations (15) and (17) are modeled identically as in Equation (10).

### 2.3. Anaplerotic Pathway and PEP Cycle

The PEP cycle with the net flux of PEP ($J_{\mathrm{PEP}}$) is coupled to the mitochondrial GTP cycle, which is modeled as follows:(19)$$J_{\mathrm{PEP}}=\frac{f_{\mathrm{ANA}}}{f_{\mathrm{ANA},0}}\;\left(J_{\mathrm{GTP},\mathrm{pyr},\mathrm{TCA}}+J_{\mathrm{GTP},\;\mathrm{FFA},\mathrm{TCA}}\right)\left(1-k_{\mathrm{md}}\right).$$

The ratio $\frac{f_{\mathrm{ANA}}}{f_{\mathrm{ANA},0}}$ determines the FFA-dependent increase in $J_{\mathrm{PEP}}$ (see Equation (8)). Since 1 GTP is consumed during the formation of 1 PEP, the consumption of GTP in mitochondria is also increased by this ratio. This is possible if we have a sufficient concentration of GTP molecules in the mitochondria. Because mammalian mitochondria lack a GTP transporter, GTP is effectively trapped in the mitochondrial matrix [66,67]. PEP diffuses to the plasma membrane, where it is converted to pyruvate by PK. This reaction is coupled by the non-oxidative phosphorylation of ATP that accumulates near K_ATP_ channels [33,39,46,68]. Consequently, the flux of PEP is equal to PEP-cycle-dependent ATP generation ($J_{\mathrm{ATP},\mathrm{PEP}}=J_{\mathrm{PEP}}$).

### 2.4. Mechanisms of Insulin Secretion

K_ATP_ channels play a central role in the regulation of insulin secretion. ATP directly inhibits the K_ATP_ channels by binding to Kir6.2 subunits, while ATP and ADP activate the channel by interacting with the NBFs of SUR [68]. A higher ATP/ADP ratio decreases K_ATP_ channel activity, resulting in increased Ca^2+^ concentration and exocytosis of insulin. Previously, we described in detail the role of K_ATP_ channels as a coupling step between the metabolic and electrical activities of beta cells [46]. The phenomenological relationship between K_ATP_-channel conductance and hormone secretion was modeled by fitting the results of the electrophysiological model of Pedersen et al. [69]. The same equations linking K_ATP_ channel conductance to the insulin secretion rate are also used in this model. The K_ATP_-channel conductance is modeled according to the proposal of Magnus & Keizer [70].

The concentration of cytosolic ATP, which affects K_ATP_ channel conductance, is determined by the balance between ATP production and consumption. ATP is produced by glucose metabolism and by FFA metabolism. The rate of the ATP production is given by:(20)$$J_{\mathrm{ATP},\mathrm{cyt}}=J_{\mathrm{ATP},G}+J_{\mathrm{ATP},\mathrm{FFA}}\;$$

In general, ATP hydrolysis is assumed to increase with the energy state of the cell, and the kinetics is often modeled according to Michaelis-Menten kinetics (e.g., [71]). The rate of ATP hydrolysis is described by:(21)$$\begin{array}{c} J_{\mathrm{ATPase}}=k_{\mathrm{ATPase}}\;\frac{ATP_{\mathrm{cyt}}^{2}}{K_{m,\mathrm{ATPase}}^{2}+ATP_{\mathrm{cyt}}^{2}}.\# \end{array}$$
where $k_{\mathrm{ATPase}}=350\;\mu M/s$ and $K_{m,\mathrm{ATPase}}=2000\;\mu M/s$. The concentration of $ATP_{\mathrm{cyt}}$ in the steady-state approximation when the ATP production rate, $J_{\mathrm{ATP},\mathrm{cyt}}$, is in equilibrium with the ATP hydrolysis rate $J_{\mathrm{ATPase}}$ is given by:(22)$$ATP_{\mathrm{cyt}}=k_{m,\mathrm{ATPase}}\;{\left(\frac{J_{\mathrm{ATP},\mathrm{cyt}}}{k_{\mathrm{ATPase}}-J_{\mathrm{ATP},\mathrm{cyt}}}\right)}^{\frac{1}{2}}.$$

Considering the accumulation of ATP near the cell membrane due to the PEP cycle, the ATP concentration is modeled by:(23)$$\frac{dATP}{dt}=k_{\mathrm{ATP},1}\;J_{\mathrm{PEP}}-k_{\mathrm{ATP},2}\left(ATP-ATP_{\mathrm{cyt}}\right).$$

Consequently, the ATP concentration at steady state is given by:(24)$$ATP=ATP_{\mathrm{cyt}}+\frac{k_{\mathrm{ATP},1}}{k_{\mathrm{ATP},2}}\;J_{\mathrm{PEP}}.$$

The ratio $\frac{k_{\mathrm{ATP},1}}{k_{\mathrm{ATP},2}}=200$ is chosen so that the ATP concentration range coupled to the K_ATP_ conductance [70] is consistent with experimental data [72].

## 3. Results

Using the mathematical model presented in Section 2, we investigate the mechanisms of GSIS and the role of FFA exposure in modulating the physiological beta cell response to plasma glucose. Although it is clear that acute and chronic exposure to FFAs is involved in both altered metabolic and signaling pathways, we are mainly interested in the effects of FFAs from a bioenergetic perspective. In particular, model results reveal the extent to which high lipid levels induce changes in energy production (ATP), which allows us to understand changes in insulin secretion due to cell exposure to FFAs. Furthermore, independent of the analysis of the oxidative pathways of glucose and FFAs, we highlight the essential role of anaplerotic pathways in GSIS. We focus on the PEP cycle as an important anaplerotic pathway in the beta cell that provides an additional mechanism to influence the conductance of K_ATP_ channels and the resulting insulin secretion.

Figure 2 shows the metabolic fluxes that contribute to ATP production. The increase in plasma lipids and the resulting higher exposure of beta cells to FFA is modeled by increasing the parameter $f_{\mathrm{FFA}}$. Figure 2A shows the effects related to the oxidative metabolic pathways. An increase in plasma glucose concentration (G) increases the glucose oxidation pathway ($J_{\mathrm{ATP},\mathrm{GO}}$), whereas the FFA oxidation pathway ($J_{\mathrm{ATP},\;\mathrm{FFAO}}$) decreases. High glucose levels inhibit FFA oxidation by blocking the uptake of acyl-CoA into mitochondria (see Equation (13)). Increasing the parameter $f_{\mathrm{FFA}}$ has the opposite effect on the glucose and FFA oxidation pathways—as FFAs become the preferred metabolic fuel, glucose oxidation decreases at all glucose concentrations. The FFA-mediated decrease in glucose oxidation results from the reduced glycolytic flux $J_{\mathrm{ATP},\;\mathrm{gly}}$ (see Equation (1)), as shown in Figure 2B. In addition, pyruvate is diverted to the anaplerotic pathway (see Equation (8)), resulting in increased ATP production via the PEP cycle ($J_{\mathrm{ATP},\mathrm{PEP}}$), as shown in Figure 2B. The increase in $J_{\mathrm{ATP},\mathrm{PEP}}$ (Equation (19)) results from the acceleration of the mGTP cycle due to the increased entry of FFA into the TCA cycle. In addition, the availability of OAA increases with higher PC-catalyzed flux (see Figure 1).

Changes in ATP fluxes, as shown in Figure 2, result in changes in subplasmalemmal ATP concentration near K_ATP_ channels (see Equation (24)). Acute exposure of a healthy beta cell to FFAs results in increased intracellular ATP concentrations which are more pronounced at low plasma glucose concentrations, as shown in Figure 3A. Higher ATP levels at low glucose concentrations result from increased basal FFA oxidation and increased accelerated PEP cycles, as shown in Figure 2. The PEP cycle essentially produces a flux of TCA-cycle-derived GTP to energetically equivalent subplasmalemmal ATP. Both effects contribute to altered insulin secretion, as shown in Figure 3B. Basal relative insulin secretion (RIS) increases as a result of both plasma glucose concentration and increased FFA exposure.

We further address the changes in insulin secretion that result from chronic exposure of beta cells to FFAs (see Figure 4). First, we note that the increase in mitochondrial proton leak ($k_{p,\mathrm{leak}}$) through the inner mitochondrial membrane reduces glucose oxidation (Equations (9) and (10)) and FFA oxidation (Equations (15) and (17)), as shown in Figure 4A. However, increased proton flux promotes the entry of FFA into the TCA cycle (Equations (13) and (14)), which accelerates mGTP and PEP cycles (Equation (19)). As shown in Figure 4B, increased proton leak increases ATP production derived from the PEP cycle.

Changes in ATP production due to increased mitochondrial proton leak ($k_{p,\mathrm{leak}}$), as shown in Figure 4, lead to changes in insulin secretion (Figure 5). As shown in Figure 5A, mitochondrial proton leak leads to a slight increase in basal insulin secretion, whereas insulin secretion is significantly reduced at high glucose levels. The increase in basal insulin secretion results from the increased subplasmalemmal ATP levels near the K_ATP_ channels. Whereas ATP production decreases due to decreased glucose ($J_{\mathrm{ATP},\mathrm{GO}}$) and FFA ($J_{\mathrm{ATP},\mathrm{FFAO}}$) oxidation, acceleration of the PEP cycle ($J_{\mathrm{ATP},\mathrm{PEP}}$) leads to an increase in ATP flux into the subplasmalemmal space. However, at high glucose concentrations, the increase in $J_{\mathrm{ATP},\mathrm{PEP}}$ cannot compensate for the loss that results from reduced metabolite oxidation, leading to decreased insulin secretion.

For long-term exposure of beta cells to FFAs, we also consider mitochondrial dysfunction, modeled by the parameter $k_{\mathrm{md}}$. Figure 5B shows the effects of mitochondrial dysfunction on insulin secretion. Because of a decrease in all fluxes ($J_{\mathrm{ATP},\mathrm{GO}}$, $J_{\mathrm{ATP},\mathrm{FFAO}}$, $J_{\mathrm{ATP},\mathrm{PEP}}$), which are conditioned by mitochondrial function, RIS decreases.

The combination of each effect on RIS is shown in Figure 6. Short-term exposure of beta cells to FFAs, when the decrease in mitochondrial function is not yet present, increases both basal insulin secretion and GSIS (curve b). When mitochondrial proton leak is enhanced, basal insulin secretion increases slightly, whereas it decreases at high glucose (curve c). Because to mitochondrial dysfunction, insulin secretion is further reduced at all glucose concentrations (curve d), and the cell secretes approximately the same amount of insulin regardless of plasma glucose concentration, which is physiologically undesirable and can be observed in T2DM patients.

## 4. Discussion

This study provides insight into the major cellular mechanisms involved in insulin secretion. It devotes the simplistic understanding of the beta cell as the sole glucose sensor to the notion of much broader machinery for sensing glucose and other metabolites, particularly FFAs. In the model, the acute exposure of beta cells to FFAs increases insulin secretion due to increased FFA oxidation and an accelerated PEP cycle. This is consistent with experiments showing that after only 4 h, an increase in CPT 1 activity measured in isolated mitochondria leads to increased FFA oxidation, resulting in two-to-three-fold higher FFA oxidation in cells exposed to FFAs measured at low glucose compared to control [73]. Moreover, the interplay of FFA metabolism and its role in modulating the PEP cycle is consistent with a 30% increase in PC mRNA expression and thus a 2.5-fold increase in insulin secretion at 3.3 mM and a 40% increase at 27 mM during an 8-h exposure to 250 μM palmitate [74]. The same study showed that GLUT-2 mRNA expression was reduced by approximately 30% in 48 h in islet cultures with 250 μM palmitate. Furthermore, GK mRNA expression was also reduced by approximately 30%. Other studies have also shown that a high-fat diet (HFD) reduces mRNA expression of GLUT-2 and GK, diminishing glucose oxidation [75,76]. These experimental findings are consistent with our model predictions, showing reduced glycolytic flux and consequently reduced glucose oxidation during exposure to FFAs.

In contrast to acute FFA administration, chronic exposure to FFAs inhibits insulin release under glucotoxic conditions [10]. This is supported by studies showing that glucose-induced insulin release was increased in islets cultured with palmitate for 8 h, whereas glucose-induced insulin release was significantly impaired by the simultaneous presence of palmitate when the culture time was extended to 48 h [74]. There are several reasons for this. The effect of FFAs on insulin release has been associated with FFA metabolism [10,77,78] and signaling via the G protein-coupled receptor FFAR1/GPR40 [10,79,80,81]. While we are aware of their importance and impact on insulin secretion, the latter were not included in our model since they would not qualitatively influence our results but would rather amplify the individual effects. It has been shown that FFAR1 in the pancreatic beta cell plays an essential role not only in the acute potentiation of GSIS by palmitate but also in the negative long-term effects of palmitate on GSIS and insulin content [81]. Recently published research has also focused on the glycerolipid/NEFA cycle, which provides lipid signals through its lipolysis arm [20,82]. Moreover, mitochondrial dysfunction in the beta cells included in our model has been recognized as one of the most critical consequences of the chronic elevation of FFA, caused mainly by impaired peroxisome proliferator-activated receptor signaling, increased oxidative stress levels, and lipotoxic modulation of the PI3K/Akt and MAPK/ERK pathways [12].

Chronic exposure to FFAs is one of the main problems in beta cell reprogramming. In particular, the detour of oxidative processes from glucose to FFA, sparing pyruvate for anaplerotic purposes, may be one of the leading pathophysiological pathways to inappropriate energy metabolism and beta cell dysfunction, often resulting in beta cell death. The increased basal anaplerosis via the PEP cycle leads to increased basal insulin secretion and beta cell exhaustion with an inappropriate response to high glucose concentrations. The increased basal insulin secretion under lipotoxic conditions shown by our results is consistent with research showing that FFAs increase insulin secretion at fasting glucose concentrations and that improved mitochondrial metabolism is critical for this effect, which is further enhanced by FFAR1 [9]. Research has shown that insulin secretion increases approximately two-fold and continues to increase after a few days. Some studies have demonstrated increased insulin secretion at low glucose levels [83,84,85], while others have also shown a decreased insulin secretion at higher glucose concentrations [84,85], which is consistent with our findings. In particular, clonal beta cells exposed to oleate for 72 h exhibited impaired GSIS and decreased cellular ATP [84], which is consistent with our results, suggesting that the decreased insulin secretion results from lower ATP concentration due to decreased glucose oxidation and mitochondrial dysfunction. The same study also showed that mitochondria of oleate-exposed cells exhibit increased depolarization caused by acute oleate treatment, which is due to the increase in FFA transport function of UCP2. According to our results, the increased proton leak indeed increases basal insulin secretion and, most importantly, decreases GSI, indicating that chronic exposure to FFA with excessive oxidation of FFAs is a pathway to beta cell exhaustion with too high NGSIS and too low GSIS. The elevated FFAs also block the glucose entry [47], and in a vicious cycle with glucolipotoxicity, beta cells cannot efficiently sense glucose [86,87]. This is one of the most prominent pathways to the pathologies of metabolic syndrome and T2DM associated with obesity and hyperlipidemia, which are statistically observable [6] and represent an even greater problem in modern societies.

In conclusion, the computational model presented here, which incorporates all the basic anaplerotic and cataplerotic mechanisms in beta cells responsible for insulin secretion, provides quantitative estimates of the effects of short- and long-term exposure of beta cells to elevated concentrations of FFA. The results indicate that the major trigger for insulin secretion is the anaplerotic pathway via the phosphoenolpyruvate cycle, which is impaired by FFA and is particularly destructive during long-term chronic exposure to FFA, resulting in increased insulin secretion at low blood glucose and inadequate insulin secretion at high blood glucose. The observed FFA-induced disruption of anaplerotic pathways is consistent with the physiological role of insulin as one of the major anabolic hormones. Future studies could extend the model to include additional signaling pathways and the role of amino acids as metabolites and signaling molecules in the mechanisms of beta cell function and insulin secretion.

## Figures and Tables

**Figure 1 biomedicines-10-01627-f001:**
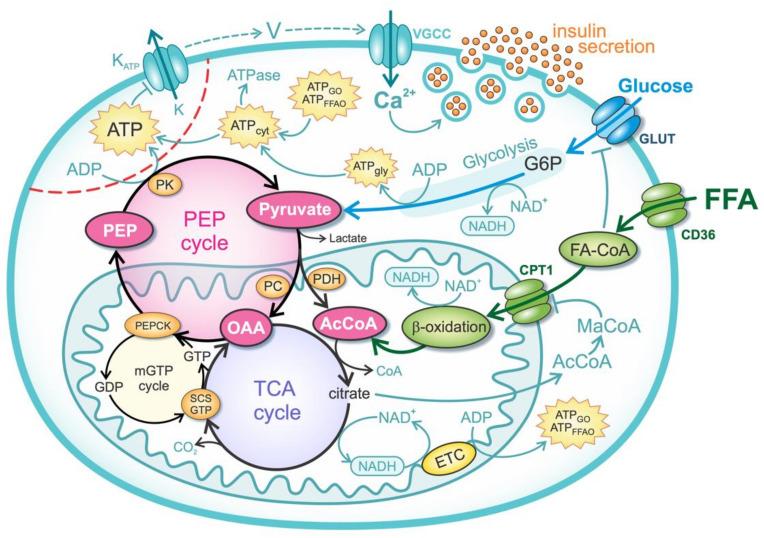
Schematic representation of a mathematical model. ATP—ATP concentration near K_ATP_ channels, ATP_cyt_—cytosolic ATP concentration, ATP_gly_—net glycolytic ATP production, ATP_FFAO_—total ATP production by FFA metabolism, ATP_GO_—ATP production by oxidation of pyruvate and reducing equivalents from glycolysis, AcCoA—acetyl-coenzyme A, CD36—transmembrane FFA transport protein (FFA translocase), CPT1—carnitine palmitoyltransferase 1, ETC—electron transfer chain, FA-CoA—fatty-acyl-coenzyme A, GLUT—glucose transporter, G6P—glucose 6-phosphatase, MaCoA—malonyl-coenzyme A, OAA—oxaloacetate, PC—pyruvate carboxylase, PDH—pyruvate dehydrogenase, PEP—phosphoenolpyruvate, PEPCK—phosphoenolpyruvate carboxykinase, PK—pyruvate kinase, SCS-GTP—GTP-specific succinyl-CoA synthetase.

**Figure 2 biomedicines-10-01627-f002:**
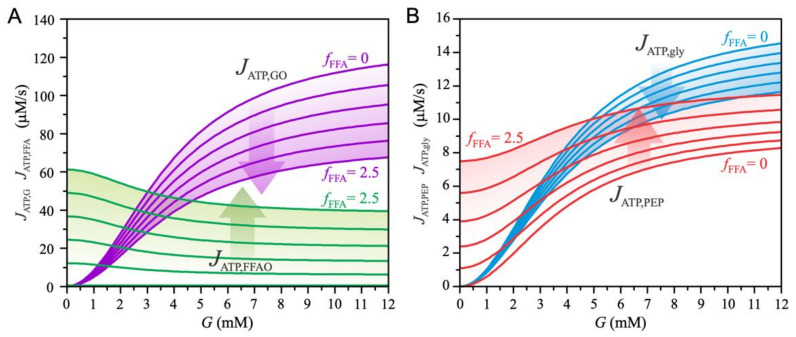
Glucose-dependent metabolic fluxes of ATP production and the effects of beta cell exposure to FFA ($f_{\mathrm{FFA}}$). The parameter $f_{\mathrm{FFA}}$ is varied between 0 and 2.5 with a step size of 0.5. (**A**) ATP production due to glucose oxidation ($J_{\mathrm{ATP},\mathrm{GO}}$) and ATP production due to FFA oxidation ($J_{\mathrm{ATP},\mathrm{FFAO}}$). (**B**) ATP production due to the PEP cycle ($J_{\mathrm{ATP},\mathrm{PEP}}$) and due to glycolysis ($J_{\mathrm{ATP},\mathrm{gly}}$).

**Figure 3 biomedicines-10-01627-f003:**
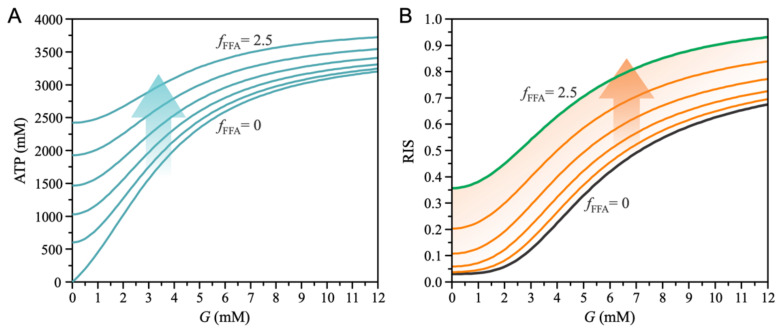
Effect of FFAs ($f_{\mathrm{FFA}}$) on ATP concentration near K_ATP_ channels. The value of parameter $f_{\mathrm{FFA}}$ increases from 0 to 2.5 with a step size of 0.5. (**A**) Glucose-dependent ATP concentration. Exposure to FFA increases basal ATP concentration, especially at low glucose levels, which consequently decreases glucose dependence. (**B**) Glucose-dependent relative insulin secretion (RIS). The FFA exposure increases basal RIS.

**Figure 4 biomedicines-10-01627-f004:**
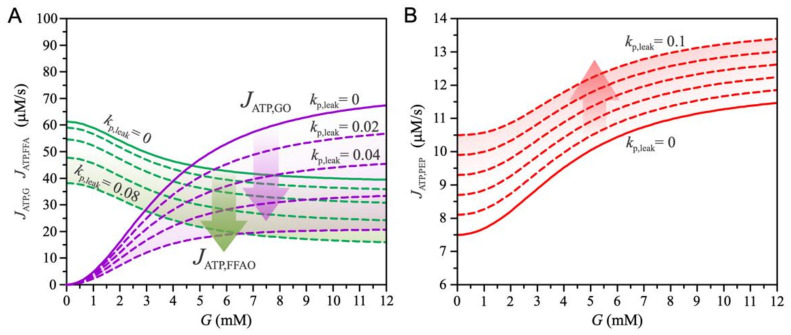
The effects of increased mitochondrial proton leak ($k_{p,\mathrm{leak}}$) on energy metabolism ($f_{\mathrm{FFA}}=2.5$ (see Figure 1)). The value of parameter $k_{p,\mathrm{leak}}$ increases with a step size of 0.02. (**A**) Glucose-dependent ATP production due to glucose oxidation ($J_{\mathrm{ATP},\mathrm{GO}}$) and FFA oxidation ($J_{\mathrm{ATP},\mathrm{FFAO}}$). Increased mitochondrial proton leak reduces glucose and FFA oxidation. (**B**) Glucose-dependent PEP-cycle-derived ATP production. Increased mitochondrial leak increases ATP production.

**Figure 5 biomedicines-10-01627-f005:**
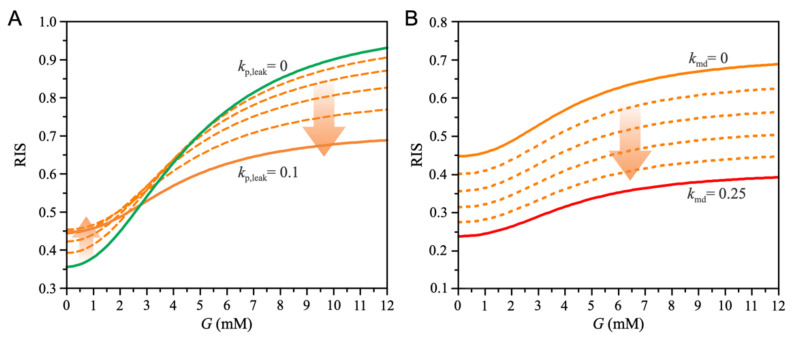
Effects of chronic lipid exposure on relative insulin secretion (RIS). (**A**) Effects of proton leak on glucose-dependent RIS. Increased proton leak flattens the glucose-dependent RIS curve, increases RIS at low glucose concentrations, and decreases RIS at high glucose concentrations. The value of parameter $k_{p,\mathrm{leak}}$ increases from 0 to 0.1 with a step size of 0.02. (**B**) The effects of mitochondrial dysfunction on glucose-dependent RIS. Increased mitochondrial dysfunction reduces RIS at all glucose concentrations. The value of parameter $k_{\mathrm{md}}$ increases from 0 to 0.25 with a step size of 0.05.

**Figure 6 biomedicines-10-01627-f006:**
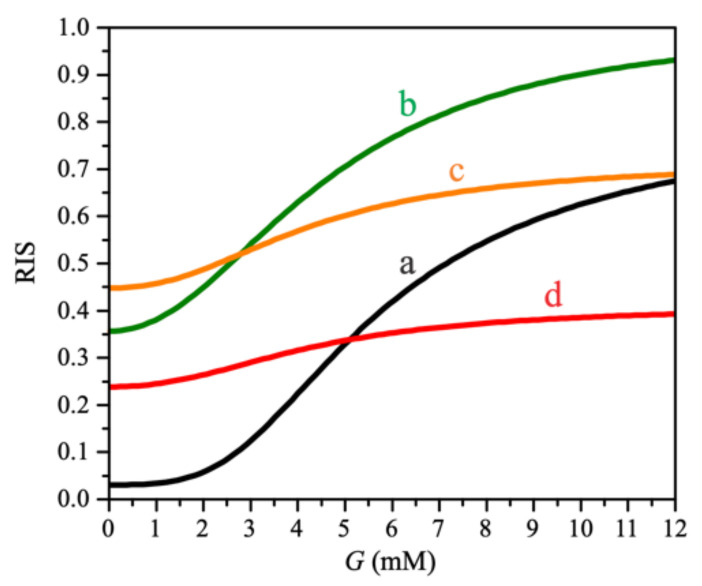
Influence of individual factors on glucose-dependent relative insulin secretion (RIS). (**a**) Absence of FFAs ($f_{\mathrm{FFA}}=0$, $k_{p,\mathrm{leak}}=0$, $k_{\mathrm{md}}=0$). A response of a healthy beta cell under physiological conditions. (**b**) The presence of FFAs, without increase in mitochondrial proton leak and mitochondrial dysfunction ($f_{\mathrm{FFA}}=2.5$, $k_{p,\mathrm{leak}}=0$, $k_{\mathrm{md}}=0$). RIS is increased at all glucose concentrations. (**c**) Presence of FFAs and increased proton flux, without mitochondrial dysfunction ($f_{\mathrm{FFA}}=2.5$, $k_{p,\mathrm{leak}}=0.1$, $k_{\mathrm{md}}=0$). The RIS curve is flattened. (**d**) Presence of FFAs with increased mitochondrial proton leak and mitochondrial dysfunction ($f_{\mathrm{FFA}}=2.5$, $k_{p,\mathrm{leak}}=0.1$, $k_{\mathrm{md}}=0.25$). RIS is reduced at all glucose concentrations.

## Data Availability

Not applicable.

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
