# Peer review of "Lipotoxicity in a Vicious Cycle of Pancreatic Beta Cell Exhaustion"

_biomedicines, 2022, doi:10.3390/biomedicines10071627_

Round 1
Reviewer 1 Report
Title : Lipotoxicity in a Vicious Cycle of Pancreatic Beta Cell Exhaus tion
This study evaluated, by computational model to quantitatively estimate, the effec of free fatty acid on insulin secretion and blood glucose level. The model in this study showed a possible lipid induced disruption of anaplerotic pathways. In my opinion it is a good study and can accpeted for publication in this form
Author Response
Response to Reviewer 1 Comments
This study evaluated, by computational model to quantitatively estimate, the effec of free fatty acid on insulin secretion and blood glucose level. The model in this study showed a possible lipid induced disruption of anaplerotic pathways. In my opinion it is a good study and can accpeted for publication in this form.
Response: Thank you for the positive evaluation of our work.
Reviewer 2 Report
The authors addressed a very interesting and newly emerging issue, that is the role of increased levels of free fatty acids on insulin secretion by pancreatic beta cells. In particular, the authors developed a computational model useful for the quantitative evaluation of the effects of specific metabolic pathways on insulin secretion and for the assessment of the effects of short- and long-term exposure of pancreatic beta cells to elevated concentrations of free fatty acids.
Results from this computational analysis show that the major trigger for insulin secretion is the anaplerotic pathway via the PEP cycle, which is affected by free fatty acids via UCP2 and H+ leak and has dramatically negative effects in long-term chronic hyperlipidemia,
The paper is interesting and informative. However, some points needs to be addressed to improve the manuscript quality.
11. The reference list should be updated with more recent papers discussing the effects of increased free fatty acids on insulin secretion as well as papers reporting the role of PEP pathway and UCP2 in hyperlipidemia-induced effects. Major findings of these papers should be commented in the Discussion.
2. Please provide better quality figures, with larger characters labelling the curves of the graphics
Reviewer 3 Report
This paper is potentially of interest to others in the field. However, there are some matters arising:
1. The assumptions made in developing the model - surely they should be assessed in a single cell or tissue type, from the same species?
2. Throughout the manuscript, the authors need to define abbreviations at first usage; they should also not be present in the Abstract
3. The authors need to remove all non- standard and colloquial terms, such as 'high revs', 'overheated', 'blind' 'Swiss army knife' and so on - they don't contribute to clarity.
4. In the section on FFA, the authors don't discriminate the impact of differing FFA species; for example, the impact of palmitate versus longer chain fatty acids
5. Query the use of 'impending' in paragraph 1 - surely the authors intend preceding?
6. The authors should take care to explain key terms when they arise in the text.
7. There is a degree of repetition in the text which should be eliminated.
Reviewer 4 Report
Major comment(s):
This article is rather hypothesis generating including many mathematical formulas/computational models. It is not clear if it is review or original article - seems to be mixture.
The term dyslipidemia does not correspond with free fatty acids (FFA) and this should be taken into account already in abstract.
I strongly recommend to focus on maximally one aspect - for example - chronic effect of FFA on pancreatic beta-cells, and use FFA this term throughout the manuscript (sometimes there is used term lipids, which could be confusing - for example rr. 286, 378 …).
There are repetitive sentences - r. 482: “We show that acute exposure of beta cells to FFA increases insulin secretion” vs. r. 498 “Our results are well consistent with the fact that acute administration of fatty acids increases insulin release from beta cells“ - these facts for the long time and well known as the fact that chronic exposure to FFA decreases secretion of insulin from beta-cells.
In addition, there are many types of (circulating) FFA saturated, monounsaturated, polyunsaturated, omega 3, 6 and this should be taken into account not talking about those bind in triacylglycerols. This should be made clear.
Minor comments/suggestions:
I miss list of abbreviations which could be very valuable.
Also recommend to refer to (and read) older article with still some valid ideas:Tan MH, Havel RJ, Gerich JE, Soeldner JS, Kane JP. Pancreatic alpha and beta cell function in familial dysbetalipoproteinemia. Horm Metab Res. 1980:421-5. doi: 10.1055/s-2007-999165. PMID: 7000652.
In summary: In short, this article is too narrative and should be restructured/shortened and more focused on the main idea - lipotoxicity- FFA and beta cells- in computational model(s).
Round 2
Reviewer 3 Report
The authors have addressed the issues raised. However, I emphasize again, the methodology employed is not within my purview.
Reviewer 4 Report
In general, safisfactory changes were made. Just two minor points:_
1. r. 273 instead of TAG are converted to Lc_FFA consider TAG are split to ...
2. at the end of the article/Discussion Conclusion should be highlighted/separated to be seen in the text.
